# Definition of Exergetic Efficiency in the Main and Emerging Thermal Desalination Technologies: A Proposal

Nenna Arakcheeva El Kori *,†, Ana M. Blanco-Marigorta †🆔 and Noemi Melián Martel †🆔

Department of Process Engineering, Industrial and Civil Engineering School, University of Las Palmas de Gran Canaria, Campus de Tafira s/n, 35017 Las Palmas de Gran Canaria, Spain; anamaria.blanco@ulpgc.es (A.M.B.-M.); noemi.melian@ulpgc.es (N.M.M.)
* Correspondence: nenna.el101@alu.ulpgc.es
† These authors contributed equally to this work.

**Abstract:** Increasing attention is being given to reduce the specific energy consumption in desalination processes, which translates into greater use of exergy analysis. An exergetic analysis provides relevant information related to the influence of the efficiency of a single component in the global plant performance and in the exergy cost of the product. Therefore, an exergy analysis identifies the main improvement potentials in a productive thermodynamic process. Related to desalination technologies, many previous papers deal with the calculation of the parameters involved in the exergy analysis, the exergetic efficiency of different processes, plants, and technologies among them. However, different approaches for formulating the exergetic efficiency have been suggested in the literature, often without sufficient understanding and consistency. In this work, these formulations, applied to the main desalination components and processes, are compared and critically reviewed. Two definitions of exergy efficiency are applied to the desalination components of the three main thermal desalination processes (multieffect distillation–thermal vapour compression, multistage flash distillation, and direct-contact membrane distillation). The results obtained for the exergy efficiency of the MED-TVC, MSF, and DCMD processes for the input–output approach are 21.35%, 17.08%, and 1.28%, respectively, compared to the consumed–produced approach that presented 3.1%, 1.58%, and 0.37%, respectively. The consumed–produced approach seems to better fit the thermodynamic behaviour of thermal desalination systems.

**Keywords:** thermal desalination; water–energy nexus; exergy efficiency; exergy

## 1. Introduction

Freshwater is a scarce natural resource essential for human life. The demand for freshwater has increased sixfold over the last 100 years, steadily increasing by about 1 percent per year. This situation is even more alarming in areas where natural water resources are scarce or nonexistent. Currently, about 3.6 billion people live in areas at risk of water stress for at least one month a year, and the population could reach between 4.8 and 5.7 billion by 2050 [1]. By 2030, a global water deficit of 40% is expected, which will be aggravated by climate change, changes in consumption patterns caused by economic development, and population growth. This means that conventional water sources such as lakes, rivers, or aquifers are no longer able to meet the water demand in many regions of the world. For this reason, obtaining freshwater is considered one of the main challenges at the global level, included in the Sustainable Development Goal (SDG) 6, one of the 17 goals established in 2015 by the United Nations General Assembly, aimed at ensuring access to clean water for current and future generations [1,2] Because 97.5% of surface water is saline, seawater desalination processes have positioned themselves as a high-value alternative for freshwater production, especially in areas where natural water resources are not capable of meeting water demand [3]. Currently, there are nearly 16,000 operational desalination

plants, with a daily production of approximately 95.37 million m$^3$/day. Seawater is most frequently used in desalination processes, accounting for 61% of the water produced, followed by brackish water and river water with 21% and 8% of the water produced, respectively, [4]. Thermal processes such as multieffect distillation (MED) or multistage flash distillation (MSF) are responsible for the production of 18% and 7% of desalinated water worldwide [5]. However, these technologies are considered as intensive in energy consumption, using both thermal energy as the primary energy to produce the phase change and electrical energy for operating conditions. Table 1 summarises the energy consumption of the main thermal-based and membrane-based desalination technologies.

**Table 1.** Specific energy consumption for the main thermal-based and membrane-based desalination technologies. Adapted from [6,7].

| Technology | Electrical Energy (kWh/m$^3$) | Thermal Energy (kWh/m$^3$) | Total Energy Consumption (kWh/m$^3$) |
|---|---|---|---|
| MSF | 2.5–5 | 40–120 | 21–59 |
| MED | 2–2.5 | 30–120 | 15–57 |
| MD | 1.5–4 | 4–40 | 3–22 |
| SWRO | 3–6 | - | 3–6 |
| ED | 1–3.5 | - | 1–3.5 |
| BWRO | 0.5–3 | - | 0.5–3 |

Exergy analysis is a tool recognised worldwide for its use in assessing the thermodynamic performance (development, evaluation, and improvement) of an energy-conversion system by identifying the location, cause, and estimation of thermodynamic inefficiencies. Its implementation in the industry field is still very low, due in part to a lack of standardisation of definitions, leading to mixed results [8]. Different approaches for formulating the exergetic efficiency can be found in the literature [9–15], which can be classified into two main categories: (a) total exergy efficiency or input–output and (b) consumed–produced exergy efficiency. Brodyansky et al. [11] offered a deep analysis of the terminology issue and how these terms can lead to different results and formulations for exergy efficiency by using different examples. Sciubba et al. [16] categorised exergy efficiencies into three general definitions. Lior and Zhang [17] classified and made distinctions between different definitions for energy, exergy, and second-law efficiencies. They referred to the input–output efficiency as the total or overall efficiency and suggested its use for processes where a major part of the output is "useful". Cornelissen [18] made a comparison between three exergy efficiency definitions. The terms product exergy and fuel or consumption exergy, from the consumed–produced exergy efficiency category, have been extensively used in the literature by several authors, with slight differences [9–12,19,20].

In this work, two definitions of exergy efficiency are applied to the desalination components of the three main thermal desalination processes. Two of the processes represent the main conventional thermal desalination systems, and the last is one of the most researched thermal emerging technologies [21]. The key point is the definition of the exergy of the product (the exergy produced by the system) and the exergy of the fuel (the resources expended to generate the product). In desalination technologies, where the separation of salt takes place, the chemical exergy of the material streams plays also an important role and must be considered in the definition of the exergetic parameters. This work proposes the definitions of the product, the fuel, and, consequently, the exergetic efficiency of the main thermal desalination processes.

## 2. Literature Review

There are several exergy analysis models used in the literature for desalination and water purification systems. Fitzsimons et al. [22] discussed the main exergy analysis approaches, with a special focus on chemical exergy, and classified exergy equations based

on the modelling of aqueous solutions and the exergy rate calculation. The results showed significant differences between the various models and the need for a correct selection of the most appropriate model for each case.

Sharqawy et al. [23] proposed equations for thermodynamic properties of seawater and evaluated their performance compared to the ideal mixture model proposed by Cerci et al. [24]. The study proved that flow exergies are always positive except in cases where the pressure is lower than the dead state, in contrast to the results often presented using the idea -mixture model.

The chemical exergy rate calculation models and exergy efficiency estimation equations used in the most representative studies of MED, MSF, and MD thermal processes are presented below.

### 2.1. Multieffect Distillation

Table 2 presents the main studies related to the application of exergy analysis, with special consideration of the chemical exergy and exergy efficiency of the MED systems.

García and Gómez [25] performed an exergy analysis for a solar MED plant that achieved a system exergy efficiency of 25.7% with a fuel–product approach through the implementation of a heat recovery system.

Piacentino et al. [26] performed a detailed analysis of the exergy flows of an eight-effect MED plant. The two main approaches were performed to find the exergy efficiency of the process. Furthermore, in this case, chemical exergy was considered, assuming the ideal solutions model proposed by Cerci et al. [24]. The results showed a considerable drop in exergy efficiency values from the more generic to the more specific approach, from values of over 88% to maximum values of 7%. In addition, the lack of purpose of the exergy efficiency of the process from the input–output perspective reduced the significance of the result, suggesting the result of the consumed fuel–useful product approach as the most realistic one. The result for the efficiency of the entire plant remained low because only a small part of the thermal exergy used is converted into the chemical exergy of the distillate and concentrated brine.

Carballo et al. [27] also performed a comparison of the input–output and fuel–product approaches for a 14-effect MED system, using the functions proposed by Sharqawy et al. [23] for the determination of the thermodynamic properties of seawater. In addition, only the chemical exergy associated with the distillate and process brine was considered as product exergy. The results revealed a large difference between both exergy efficiency approaches, surpassing in all scenarios 66% efficiency for the input–output approach and not reaching 1% for the fuel–product efficiency.

The MED process was also analysed in combination with processes such as thermal vapour compression and multistage flash distillation. Elsayed et al. [28] performed a MED-TVC system exergy analysis using the Sharqawy functions for the thermodynamic properties of seawater. Exergy efficiency was determined using the input–output approach. Low exergy efficiency values were observed in the study, with a range between 4 and 4.4%, with the TVC system being responsible for almost 60% of all the exergy destroyed within the system. Similarly, Moghimi et al. [29] obtained a maximum value of 3% exergy efficiency with a fuel–product approach for a five-effect MED-TVC plant with the ionic solutions model developed by Drioli et al. [30].

Eldean and Soliman [31] analysed three different scenarios for a hybrid 12-effect MSF-MED system coupled with a source of waste gases and power generation cycles. The ionic solutions model proposed by Drioli et al. for exergy flows and a inlet–outlet exergy efficiency approach were selected for the analysis. The results showed an efficiency of 62.73% for the best scenario, which produces both electricity and desalinated water, where the large production of electricity has a significant influence on the exergy efficiency of the process.

**Table 2.** Exergy analysis in multieffect distillation.

| Type of Process | Operating Conditions | Chemical Exergy | Exergy Efficiency | Ref. |
|---|---|---|---|---|
| MED | Seawater (37,000 ppm) Temperature (°C): 75 Effects (n°): 8 | $w_{min,0} = \varphi R T x_{s,feed}^{no\ dissoc}$ $w_{min} = \varphi R T x_{s,feed}^{no\ dissoc} \frac{x_{s,brine}^{no\ dissoc}}{x_{s,brine}^{no\ dissoc} - x_{s,feed}^{no\ dissoc}} ln \frac{x_{s,brine}^{no\ dissoc}}{x_{s,feed}^{no\ dissoc}}$ | $\eta_e = \frac{\sum E_{useful\ product}}{\sum E_{consumption}} = 7.5\%$ | [26] |
| MSF-MED | Seawater (45,000 ppm) Temperature (°C): 60 Effects (n°): 12 | $E^{CH} = \dot{m}[N_m R T_0(-x_w log x_w - x_s log x_s)]$ | $\eta_e = \frac{\sum E_{out}}{\sum E_{in}} = 62.7\%$ | [32] |
| MED | Seawater Temperature (°C): 66 Effects (n°): 14 | $E^{CH} = \dot{m} \sum_{i=1}^{n} w_i(u_i - u_{i0})$ | $\eta_e = \frac{\sum E_{out}}{\sum E_{in}} = 97\%$ $\eta_e = \frac{\dot{E}_P}{\dot{E}_F} = 0.2\%$ | [27] |
| MED-TVC | Seawater (42,000 ppm) Temperature (°C): 92 Effects (n°): 6 | $E^{CH} = -N_m R T_0[(-x_w ln x_w - x_S ln x_s)]$ | $\eta_e = \frac{\dot{E}_P}{\dot{E}_F} = 0.1\%$ | [33] |
| MED | Seawater (36,000 ppm) Temperature (°C): 60 Effects (n°): 5 | $E^{CH} = -N_m R T_0[(x_w ln x_w + x_S ln x_s)]$ | $\eta_e = \frac{\dot{E}x_{V,tot} - \dot{E}x_{C,in}}{\dot{E}x_{Steam,in} - \dot{E}x_{Steam,out}} = 3.1\%$ | [29] |
| MED-TVC | Seawater (40,000 ppm) Temperature (°C): 60 Effects (n°): 12 | $E_{prod}^{CH} = \varphi \dot{N}_m R T_0 X_{s,feed}^{no\ dissoc}$ | $\eta_e = \frac{\dot{W}_{net} + \dot{E}_{prod}^{CH}}{\dot{E}_{solar}}$ | [34] |
| MED-TVC | Seawater (45,000 ppm) Temperature (°C): 65 Effects (n°): 4 | $E^{CH} = \dot{m} \sum_{i=1}^{n} x_i(u_i - u_{i0})$ | $\eta_e = \frac{E_{out}}{E_{in}} = 4.4\%$ | [28] |
| MED-MCV | Seawater (42,000 ppm) Temperature (°C): 100 Effects (n°): 10 | $E^{CH} = \dot{m} n_m R T_0[-X_w(ln X_w - X_s)ln X_w]$ | $\eta_e = \frac{\dot{E}_P}{\dot{E}_F} = 4.8\%$ | [35] |
| MED-TVC | Seawater (45,000 ppm) Temperature (°C): 60 Effects (n°): 4 | $E^{CH} = \dot{m} \sum_{i=1}^{n} x_i(u_i - u_{i0})$ | $\eta_e = \frac{E_P}{E_F}$ | [36] |

*2.2. Multistage Flash Distillation*

Table 3 presents the main studies related to the application of exergy analysis with special consideration of the chemical exergy and exergy efficiency of MSF systems. Kahraman and Cengel [37] analysed thermodynamically a 22-stage MSF plant using the ideal solutions model developed by Cerci et al. [24]. The exergy efficiency was estimated as the ratio of the minimum work required for the separation process to the total exergy input. The exergy efficiency obtained was 4.2% for the entire plant, with the distillation units as the main contributors to the exergy destruction of the plant. Nafey et al. [38] obtained similar findings for exergy efficiency values below 2% by using the fuel–product approach and the ideal solutions model developed by Cerci et al. [24] for a 20-stage MSF-BR plant. In this case, the flash chambers were responsible for the majority of the exergy destroyed as well. Mabrouk et al. [39] proposed a new configuration by incorporating an MVC system into the previous MSF process, resulting in a 39% increase in exergy efficiency.

**Table 3.** Exergy analysis in multistage flash distillation.

| Type of Process | Operating Conditions | Chemical Exergy | Exergy Efficiency | Ref. |
|---|---|---|---|---|
| MSF | Seawater (46,500) <br> Temperature (°C): 90 <br> Stages (n°): 22 | $E^{CH} = -N_m RT_0 [(-x_w ln x_w - x_S ln x_s)]$ | $\eta_{e,total} = \frac{W_{min}}{\sum \dot{E}_{in}} = 4.2\%$ | [37] |
| MSF-BR | Seawater (45,000) <br> Temperature (°C): 109 <br> Stages (n°): 21 | $E^{CH} = -N_m RT_0 [(-x_w ln x_w - x_S ln x_s)]$ | $\eta_e = \frac{\dot{E}_P}{\dot{E}_F} = 1.87\%$ | [38] |
| MSF-MVC | Seawater (48,620) <br> Temperature (°C): 110 <br> Stages (n°): 20 | $E^{CH} = -N_m RT_0 [(-x_w ln x_w - x_S ln x_s)]$ | $\eta_e = \frac{\dot{E}_P}{\dot{E}_F} = 2\%$ | [39] |
| MSF | Seawater (45,000) <br> Temperature (°C): 114 <br> Stages (n°): 28 | $E^{CH} = \dot{m} \sum_{i=1}^{n} w_i (u_i - u_{i0})$ | $\eta_{e,total} = \frac{W_{min}}{\sum \dot{E}_{in}} = 5.8\%$ | [40] |
| MSF | Seawater (45,000 ppm) <br> Temperature (°C): 110 <br> Stages (n°): 28 | - | $\eta_{e,total} = \frac{W_{min}}{\sum \dot{E}_{in}} = 3\%$ | [41] |
| MSF | Seawater (44,000 ppm) <br> Temperature (°C): 90 <br> Stages (n°): 24 | $E^{CH} = \dot{m} \sum_{i=1}^{n} w_i (u_i - u_{i0})$ | $\eta_{e,total} = \frac{W_{min}}{\sum \dot{E}_{in}}$ | [42] |
| MSF-TVC | Wastewater (51,400 ppm) <br> Temperature (°C): 115 <br> Stages (n°): 25 | - | $\eta_e = \frac{\sum \dot{E}_{out}}{\sum \dot{E}_{in}} = 61.6\%$ | [43] |
| MED-MSF | Seawater (35,000 ppm) <br> Temperature (°C): 70–80 <br> Stages (n°): 14 + 14 | $E^{CH} = -N_m RT_0 [(x_w ln x_w + x_S ln x_s)]$ | $\eta_e = \frac{\sum \dot{E}_{out}}{\sum \dot{E}_{in}} = 70\%$ | [44] |
| MSF | Seawater (45,000 ppm) <br> Temperature (°C): 90–110 <br> Stages (n°): 23–26 | $E^{CH} = \dot{m} \sum_{i=1}^{n} w_i (u_i - u_{i0})$ | $\eta_{e,total} = \frac{W_{min}}{\sum \dot{E}_{in}} = 5.4 - 6.9\%$ | [45] |

Weshahi et al. [40] analysed from the exergetic perspective an existing MSF desalination plant by using the Sharqawy et al. [23] functions for the thermodynamic properties of seawater and the ratio of the minimum work of separation to the exergy of the inputs as the exergy efficiency approach. The exergy efficiency obtained was 5.8%, and the major contribution to exergy destruction belonged to the distillation units related to heat recovery. An improvement to the system through the recovery of the hot distillate water from the first stages of the process was proposed, which could improve the exergy efficiency by up to 14%.

Ghamdi et al. [41] performed an exergy analysis of a real 28-stage MSF distillation plant. In this case, chemical exergy was not considered, and the ratio of the minimum work of separation to the total exergy of inputs was taken into account for the exergy efficiency approach. Some streams revealed negative exergy values, which was explained as a consequence of the large difference in salinity between some locations in the system and the dead state. The results related to the overall exergy efficiency showed the same trend as in previous studies, with values around 3%.

### 2.3. Membrane Distillation

Table 4 presents the main studies related to the application of exergy analysis with special consideration of the chemical exergy and exergy efficiency of MSF systems.

Banat et al. [46] performed an exergy analysis of a compact and a large solar-powered AGMD system. The ideal solutions model proposed by Cerci et al. [24] and the exergy

efficiency defined as the ratio of outputs to inputs were selected. As in other cases previously mentioned, negative exergy values were found in the brine streams. The exergy efficiency of the compact and large systems were 0.01% and 0.05%, respectively. The low efficiency obtained was mainly attributed to the low distillate production of the membrane modules. Similar exergy efficiency results were presented by Mrayed et al. [47] for an experimental DCMD system using the same exergy model and efficiency approach. Two efficiency improvement options were suggested: the use of wasted energy from energy production processes and the incorporation of renewable energy. In both cases, exergy efficiency improved to 17%.

**Table 4.** Exergy Analysis in Membrane Distillation.

| Type of Process | Operating Conditions | Chemical Exergy | Exergy Efficiency | Reference |
|---|---|---|---|---|
| AGMD | Seawater (40,000 ppm) Temperature (°C): 66–35 Flux (kg/s): $3.62 \times 10^{-3}$ | $E^{CH} = -N_m R T_0 [(-x_w ln x_w - x_S ln x_s)]$ | $\eta_e = \frac{\dot{W}_{min}}{\dot{E}_{sun}} = 0.5\%$ | [46] |
| DCMD | Seawater (35,000 ppm) Temperature (°C): 70–15 Flux (kg/s): 0.027 | $E^{CH} = -\dot{m}(n_w R T_0 ln x_w)$ | $\eta_e = \frac{\sum E_{out}}{\sum E_{in}} = 28.3\%$ | [48] |
| DCMD | Seawater (30,000 ppm) Temperature (°C): 65–22 Flux (kg/s): 0.002 | $e^{CH} = \frac{RT_0}{MW}[(x_s ln x_s + x_w ln x_w) - (x_s ln x_s + x_w ln x_w)_0]$ | $\eta_e = \frac{\dot{W}_{min}}{\sum \dot{E}_{in}} = 0.03\%$ | [47] |
| VMD | Seawater (35,000 ppm) Temperature (°C): 80–77 Flux (kg/s): 0.02 | $E^{CH} = \dot{m} \sum_{i=1}^{n} w_i(u_i - u_{i0})$ | $\eta_{e,total} = \frac{W_{min}}{\sum E_{in}} = 3.3\%$ $\eta_{e,functional} = \frac{E_{evap}}{E_{in}} = 9.9\%$ | [49] |
| DCMD | Seawater (37,000 ppm) Temperature (°C): 70–31 Flux (kg/s): $0.3 \times 10^{-3}$ | - | $\eta_{e,universal} = \frac{E_{out}}{E_{in}}$ $\eta_{e,functional} = \frac{E_P}{E_F}$ | [50] |
| DCMD | Tap water (2000 ppm) Temperature (°C): 80–25 Flux (kg/s): $0.3 \times 10^{-3}$ | - | $\eta_{e,universal} = \frac{E_{out}}{E_{in}} = 85\%$ $\eta_{e,universal} = \frac{E_{out}}{E_{in}} = 70\%$ | [51] |
| AGMD | Seawater Temperature (°C): 55–27 Flux (kg/s): $1.23 \times 10^{-3}$ | $E^{CH} = -\dot{m}(n_w R T_0 ln x_w)$ | $\eta_e = \frac{E_{out}}{E_{in}} = 0.38\%$ | [52] |
| AGMD | Seawater (35,000 ppm) Temperature (°C): 62–29 Flux (kg/s): $3 \times 10^{-3}$ | $E^{CH} = \dot{m} \sum_{i=1}^{n} w_i(u_i - u_{i0})$ | $\eta_e = \frac{\dot{W}_{min}}{\dot{E}_F} = 56.3\%$ | [53] |
| DCMD | Seawater (35,000 ppm) Temperature (°C): 77–25 Flux (kg/s): 0.089 | $E^{CH} = \dot{m} \sum_{i=1}^{n} w_i(u_i - u_{i0})$ | $\eta_e = \frac{E_{out}}{E_{in}} = 56.71\%$ | [54] |
| MSVMD-MVC | Seawater (113,800 ppm) Temperature (°C): 50–25 Flux (kg/s): 0.061 | $E^{CH} = \dot{m} \sum_{i=1}^{n} w_i(u_i - u_{i0})$ | $\eta_e = \frac{\dot{W}_{min}}{\sum \dot{E}_{in}} = 6.85\%$ | [55] |

Al-Obaidani et al. [48] employed the ionic solutions model proposed by Drioli et al. [56] for the exergy analysis and the input–output for the exergy efficiency approach of an experimental DCMD setup. The energy efficiency obtained for the system was 25.6%, which could have been improved by the implementation of the heat recovery unit to 28.3%.

Miladi et al. [49] compared two models, the ideal solutions model (Cerci et al. [24]'s model) and the thermodynamic properties of seawater model (Sharqawy et al. [23]'s model), by applying them to a solar-powered VMD system. Two different approaches

for exergy efficiency were used, the overall exergy efficiency related to the input–output method and the utilitarian exergy efficiency, which is the ratio of the useful outputs to the required inputs. As observed in previous studies, the streams that contain higher salinity with regard to the dead state present negative chemical exergy values when using the Cerci et al. [24] model. A difference of 18% was found between the two models for the exergy of the process streams. The overall efficiency values for the Cerci et al. [24] model and the Sharqawy et al. [23] model were 3.25% and 2.30%, respectively. The utilitarian exergy efficiency was calculated independent of the models used and was found to be about 8% higher than the overall exergy efficiency. This is justified by the fact that the overall exergy efficiency takes into account all the exergy destroyed in the process as opposed to the exergy destroyed in the process.

## 3. Case Studies

Selected case studies are presented below; the schematic of the processes evaluated in the present study can be seen in Figures 1–3.

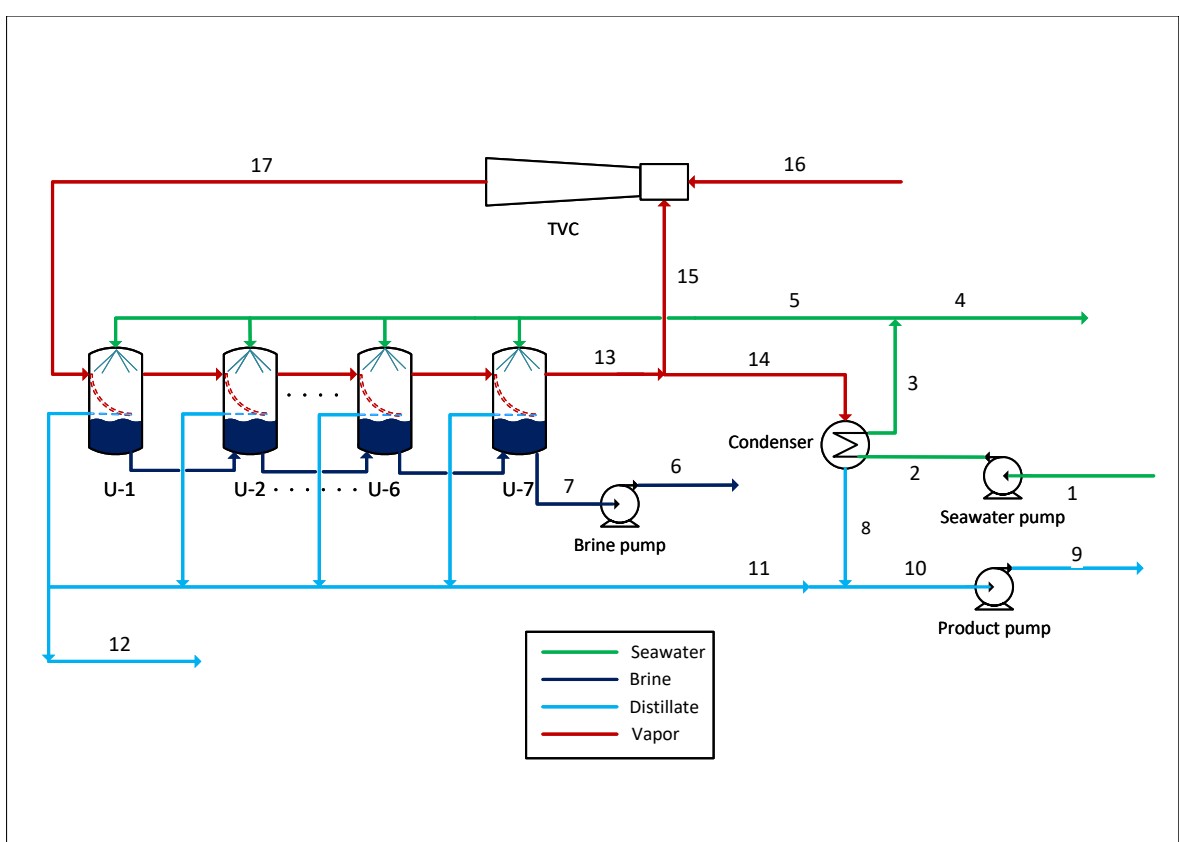

**Figure 1.** Schematic of the MED-TVC process with 7 effects.

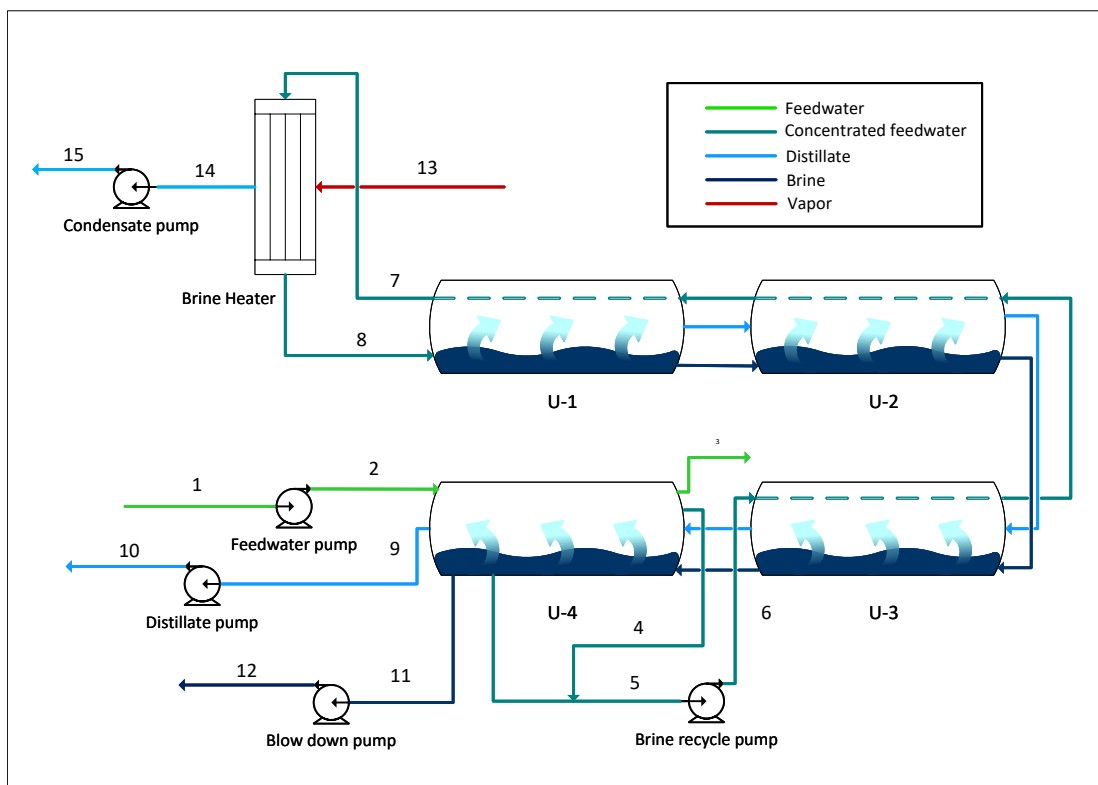

**Figure 2.** Schematic of the MSF process.

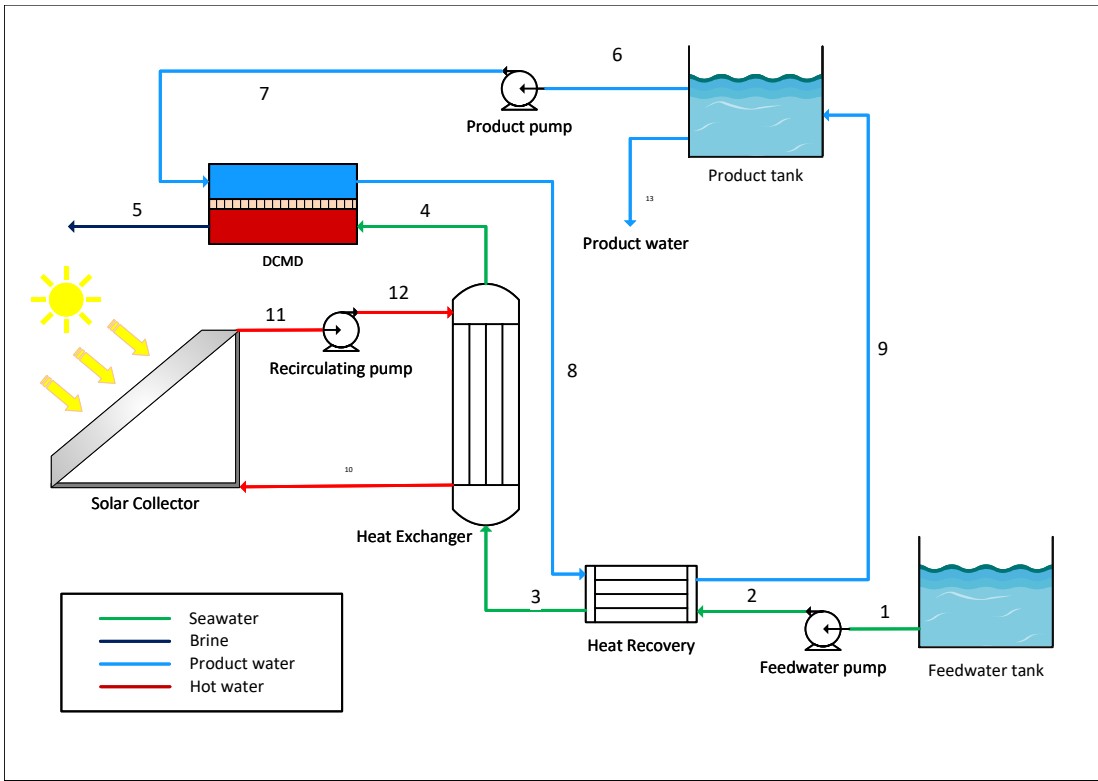

**Figure 3.** Schematic of the DCMD process.

## 3.1. MED-TVC Desalination Plant

Operating data such as temperature, pressure, flow, etc., from previous work by Eshoul et al. were used in the present study [57]. The plant, with a capacity of 24,000 m³/day,

is composed of four MED-TVC units. Each unit produces 6000 m$^3$/day and consists of seven effects. The feedwater is pumped to the condenser in order to cool the distillate product and preheat the seawater. Subsequently, part of the seawater is returned to the tank, while the rest is sprayed into each of the seven effects of the system. In turn, the TVC system provides the steam needed to generate the vapour in the first effect, which passes to the second effect where it is condensed while generating the vapour for the next effect. The condensed vapour gives rise to the distillate product of the system. For dead-state conditions, the ambient temperature $T_0$ = 20 °C, ambient pressure $p_0$ = 1.013 bar, and seawater mean salinity of 37,000 ppm were assumed.

### 3.2. MSF Desalination Plant

Operating data (temperature, pressure, flow, work input, etc.) from previous work by Khoshrou et al. were utilised in the present study [58]. The plant consists of three MSF units, each with a capacity of 180 tons per hour of distilled water. Each unit consists of 22 stages, 18 stages corresponding to heat recovery and the last 4 corresponding to heat rejection. The 22 stages are split into four desalination vessels, presented in the schematic of the process. The feedwater is initially preconcentrated in the first vessel. Some of the seawater is purged, while the rest continues through the other vessels as it is preheated and used for flash distillation. Subsequently, the preconcentrated water reaches a temperature of 115 °C in a steam-fired heater. As the water returns through the vessels, it is partially evaporated, resulting in the distillate product. For dead-state conditions, the feedwater intake parameters of temperature $T_0$ = 32 °C, pressure $p_0$ = 1.013 bar, and seawater salinity of 8.935 ppm were assumed.

### 3.3. Membrane Distillation Plant Powered by Solar Thermal System

Operating data (temperature, pressure, flow, etc.) from previous works by Okati et al. [54] and Miladi et al. [49] were used in the present study. The process is designed to treat seawater with a salinity of 35,000 ppm and produces 21.6 kg/h of product water.

The feedwater is initially preheated by the heat recovery, where the waste heat from the permeate stream is utilised. In this way, the feed stream is preheated before entering the main heat exchanger. The main heat exchanger is powered by a solar collector, which provides the operating temperature of 80 °C to the hot stream. Within the membrane module, water vapour is generated and passes through the hydrophobic membrane from the salt stream to the permeate stream due to the saturation pressure gradient generated by the temperature difference between the two streams flowing countercurrent in direct contact with the membrane surface. For dead-state conditions, the feedwater intake parameters of temperature $T_0$ = 20 °C, pressure $p_0$ = 1.01 bar, and seawater salinity of 35,000 ppm were assumed. The characteristics of the solar collector were selected from a study previously carried out by Schwantes et al. [59]. An average radiation of 800 W/m$^2$, solar collector area of 6 m$^2$, and temperature at the surface of the sun of 6000 K were considered.

## 4. Exergy Analysis

The exergetic balance of a system arises from the first and second thermodynamic laws as a combination of the energy and entropy balances, leading to determine both the quantity and quality of the resources [15]:

$$\dot{E}_Q - \dot{E}_W + \sum_j \dot{E}_{in} - \sum_j \dot{E}_{out} - \dot{E}_D = \Delta E \tag{1}$$

The total exergy of a system can be divided into four main components: physical, chemical, kinetic, and potential exergy. Being an extensive property, the specific exergy can be defined as

$$e = e^{PH} + e^{CH} + e^{KN} + e^{PT} \tag{2}$$

At rest conditions of the system with respect to the surroundings, the kinetic exergy and potential exergy terms can be neglected: $e_{KN} = e_{PT} = 0$.

Specific physical exergy values were determined by

$$e^{PH} = h - h_0 - T_0(s - s_0) \tag{3}$$

The specific chemical exergy $e^{CH}$ as well as the enthalpy, $h$, and entropy, $s$, of freshwater, brine, and seawater were calculated using the seawater thermodynamic property functions proposed by Sharqawy et al. [23].

For the exergy of the solar thermal system, the Petela expression was applied [60]:

$$\dot{E}_{solar} = A I_{solar} \left[ 1 + \frac{1}{3} \left( \frac{T_0}{T_{solar}} \right)^4 - \frac{4}{3} \left( \frac{T_0}{T_{solar}} \right) \right] \tag{4}$$

where $T_s$ is the temperature of the sun, $A$ is the area of the solar collector, $I_s$ is the solar radiation, and $T_0$ is the temperature at dead state.

*Exergy Efficiency*

Exergy efficiency quantifies the performance of an energy-dependent system from the thermodynamic point of view and is traditionally perceived as as the percentage of exergy provided to the system that can be found in the product. It is fundamental to identify both the exergy supplied to the system (exergy of the fuel) and the exergy of the product. Research studies have proposed several definitions for both terms. Some of them consider the fuel simply as all the exergy introduced into the system and the product as the exergy that leaves the system. Others relate the fuel exergy with the exergy of resources and their decrease within the system and the product as the increase of the exergy produced. Both definitions are accepted by a large number of researchers but in practice lead to different results when applied to the same processes.

Gaggioli et al. [61] referred to exergy efficiency as "true efficiency" and energy efficiency as "traditional efficiency" due to exergy efficiency's ability to locate and quantify "true" process inefficiencies in the form of exergy destruction. This general definition gives rise to the presence in the literature of various definitions of exergy efficiency, which makes it difficult to unify the term and, consequently, to obtain homogeneous results [15,61].

Definitions can be classified into two main approaches: input–output exergy efficiency and consumed–produced exergy efficiency.

Marmolejo-Correa and Gundersen [62] applied the definitions of input–output exergy and product–consumption exergy to a natural gas liquefaction process, obtaining significantly different results in both cases. A very high exergy efficiency value (98.3%) was achieved by the first definition, compared to 50.5% efficiency obtained by the second definition. The authors considered the product–consumption approach as more advanced for process evaluation.

a.   Inlet–outlet efficiency.

Kotas [63], under the name of rational efficiency, defined exergy efficiency as the ratio between the exergy transformations that constitute the output and the input of the system. Gundersen [64] defined exergy efficiency as the ratio between the useful exergy produced by the system and the total exergy of the system. Rosen and Dinçer [8] defined exergy efficiency as the ratio between the product exergy at the output of the system and the exergy at the input. Considering those definitions, the exergy balance can be expressed as

$$\sum \dot{E}_{in} = \sum \dot{E}_{out} + \dot{E}_L + \dot{E}_D \tag{5}$$

As before, the ingoing and outgoing exergy quantities may include work, heat, and cooling, with or without mass flows. The overall or total exergetic efficiency, $\eta_{e,t}$, should be, consequently, defined as

$$\eta_{e,t} = \frac{\sum \dot{E}_{out}}{\sum \dot{E}_{in}} = 1 - \frac{\dot{E}_D}{\sum \dot{E}_{in}} \tag{6}$$

b.  Consumed–produced efficiency.

The exergy supplied to the system is either delivered in the outputs or destroyed inside the system. This balance can also be expressed in terms of exergy of the fuel (the exergy of the resources), $\dot{E}_F$; exergy of the product (the desired exergy output), $\dot{E}_P$; exergy loss (with effluents), $\dot{E}_L$; or exergy destroyed (within the system due to irreversibilities), $\dot{E}_D$.

The terms product exergy and fuel or consumption exergy have been widely used in the literature by several authors, with slight differences [15,61,65–67], often indicated as the most appropriate definition for exergy efficiency. Szaargut et al. and Tsatsaronis et al. [12,14] proposed taking into account those exergy transfers that were used in the production of the desired exergy from the driving exergy of the system, for which a coherence between the purpose of the system and its exergy analysis is necessary, giving rise to the concepts of fuel exergy and product exergy. In addition, Lazaretto and Tsatsaronis [68] proposed a systematic procedure for the definition of the exergy efficiency for process components.

$$\dot{E}_F = \dot{E}_P + \dot{E}_L + \dot{E}_D \tag{7}$$

Based on the common energy efficiency definition as the ratio between the product that is obtained (work, heat, refrigeration) and the resources that must be "paid" for, the consumed–produced exergetic efficiency could be defined as the ratio between the useful exergy outputs and the paid exergy inputs [15].

$$\eta_e = \frac{desired\ output\ exergy}{input\ exergy\ required} = \frac{\dot{E}_P}{\dot{E}_F} \tag{8}$$

Both the total efficiency and the consumed–produced efficiency definitions have been applied. Table 4 presents the two main approaches for the common components of the thermal desalination processes presented in this work. The thermodynamic properties and exergy flows of the processes presented in this study were determined using the Engineering Equation Solver (EES) program.

The equations for the exergy efficiency of the components and the distillation units of the case studies are presented in Tables 5 and 6.

**Table 5.** Exergy efficiency equations for the components involved in the thermal desalination processes.

| Component | $\eta_{e,t}$ | $\eta_e$ |
|---|---|---|
| Pump | $\dfrac{\dot{E}_{out}}{\dot{E}_{in}+\dot{W}}$ | $\dfrac{\dot{E}_{out}-\dot{E}_{in}}{\dot{W}}$ |
| Heat exchanger | $\dfrac{\dot{E}_{seawater,out}+\dot{E}_{hot\ stream,out}}{\dot{E}_{seawater,in}+\dot{E}_{hot\ stream,in}}$ | $\dfrac{\dot{E}_{seawater,out}-\dot{E}_{seawater,in}}{\dot{E}_{hot\ stream,in}+\dot{E}_{hot\ stream,out}}$ |
| Condenser | $\dfrac{\dot{E}_{hot,out}+\dot{E}_{cold,out}}{\dot{E}_{hot,in}+\dot{E}_{cold,in}}$ | $\dfrac{\dot{E}_{cold,out}-\dot{E}_{cold,in}}{\dot{E}_{hot,in}-\dot{E}_{cold,out}}$ |

**Table 6.** Exergy efficiency equations for the distillation units.

| Process | $\eta_{e,t}$ | $\eta_e$ |
|---|---|---|
| MED-TVC | $\dfrac{\dot{E}_{13}+\dot{E}_7+\dot{E}_{11}+\dot{E}_{12}}{\dot{E}_{17}+\dot{E}_5}$ | $\dfrac{\dot{E}^{CH}_7+\dot{E}^{CH}_{11}+\dot{E}^{CH}_{12}+\dot{E}^{CH}_{13}-\left(\dot{E}^{CH}_5+\dot{E}^{CH}_{17}\right)}{\dot{E}^{PH}_5+\dot{E}^{PH}_{17}-\left(\dot{E}^{PH}_7+\dot{E}^{PH}_{11}+\dot{E}^{PH}_{12}+\dot{E}^{PH}_{13}\right)}$ |
| MSF | $\dfrac{\dot{E}_7+\dot{E}_9+\dot{E}_{11}}{\sum \dot{W}_{Brine}+\dot{E}_2+\dot{E}_8}$ | $\dfrac{\dot{E}^{CH}_7+\dot{E}^{CH}_{11}+\dot{E}^{CH}_9-\left(\dot{E}^{CH}_2+\dot{E}^{CH}_8\right)}{\dot{E}^{PH}_8+\dot{E}^{PH}_2-\left(\dot{E}^{PH}_7+\dot{E}^{PH}_{11}+\dot{E}^{PH}_9\right)}$ |
| DCMD | $\dfrac{\dot{E}_5+\dot{E}_8}{\dot{E}_4+\dot{E}_7}$ | $\dfrac{\dot{E}_8-\dot{E}_7+\left(\dot{E}^{CH}_5+\dot{E}^{CH}_4\right)}{\left(\dot{E}^{PH}_5+\dot{E}^{PH}_4\right)}$ |

## 5. Results and Discussion

One of the most important contributions of exergy analysis is the estimation of the contributions of system components to the overall exergy destruction, because of its ability to locate the inefficiencies present in the system components.

Figures 4–6 show the contribution of each component to the exergy destruction in the case studies.

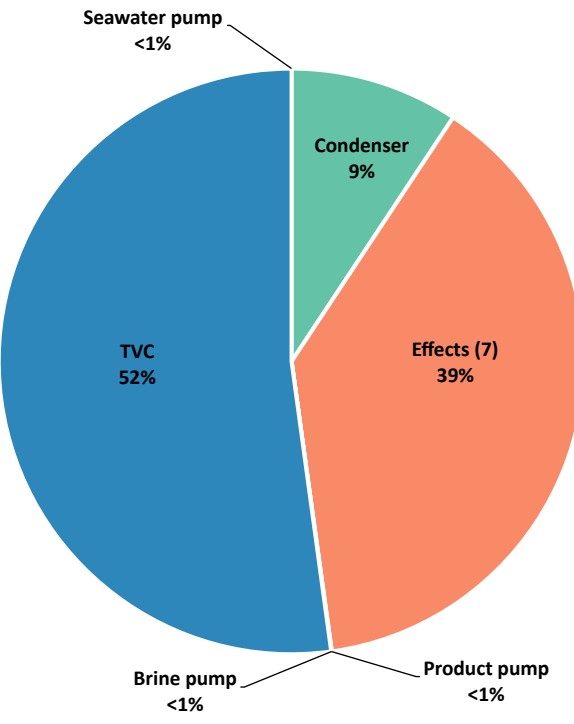

**Figure 4.** Exergy destruction by component in the MED-TVC process.

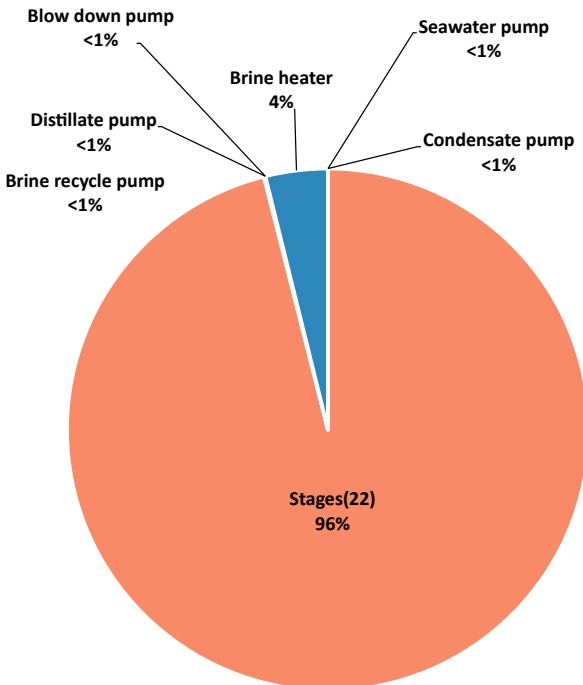

**Figure 5.** Exergy destruction by component in the MSF process.

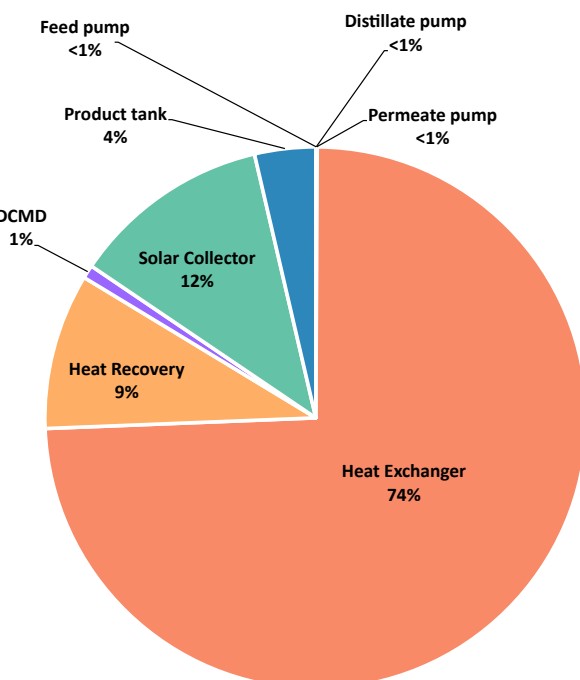

**Figure 6.** Exergy destruction by component in the DCMD process.

In the MED-TVC process, the major contribution belongs to the TVC system. As the TVC system is the main component responsible for the exergy fuel in the process, this finding is not unexpected. The second major exergy destruction occurs in the seven effects due to their responsibility for the separation of salts. The implication of the pumps in exergy destruction is almost nonexistent due to the low pressure requirements of this type of process. Eshoul et al. [57] presented a similar distribution in the contribution by component to the exergy destruction in the system, although the difference between TVC and the effects does not seem to be so pronounced. In other studies such as [36,69], the TVC source has consistently been identified as the main contributor for exergy destruction. However, in [36], the effects slightly surpassed the TVC contribution for exergy destruction.

In the MSF process, the exergy destroyed in the stages represents almost the totality of the exergy destroyed. Given the small amount of chemical exergy compared to the physical, mainly thermal, exergy required to produce it, this is a logical result. The second major exergy destruction belongs to the brine heater, which also contributes mainly in the form of thermal exergy. The distribution of exergy destruction by component presented by Khoshrou et al. [58] followed a similar pattern. However, the contribution of the stages decreased by more than 20%. Most studies present similar results, with variations mainly in the proportion of exergy destruction contributed between the stages and the brine heater [37–39,41]. As observed in the MED-TVC process, the pumps play a very small role in the exergetic destruction of the process due to the low pressure requirements, which was also concluded by Piacentino et al. [26].

The highest contribution belongs to the heat exchanger with 74%, followed by the heat recovery unit with 9%. This can be observed in other studies such as Zhu et al. [70], where the preheating system accounted for about 60% of the total exergy destroyed. As in the cases of MED-TVC and MSF, the sources of physical exergy in the form of thermal exergy, which are the main contributors to the fuel exergy in these systems, are also the main contributors to the exergy destruction in the system. As can be seen in the previous cases, the lowest contribution to the exergy destroyed corresponds to the pumps in the system since the operating pressure is close to the atmospheric pressure. Most studies are in agreement with these results [49,53,54,70,71]. The membrane module offers small contributions of only 1% to the total destroyed exergy. These values can be justified by the assumption of the pressure and temperature losses throughout the system as negligible.

Furthermore, these results are in agreement with those obtained by Mibarki et al. [53] for a system with an AGMD configuration integrated with a solar collector, operating under similar conditions of temperature and salt concentration.

The results related to the two main approaches for the exergy efficiency of the MED-TVC, MSF, and DCMD case studies are presented in Tables 7–9.

**Table 7.** Exergetic variables of the main components of the MED-TVC process.

| Component | $\eta_{e,t}(\%)$ | $\dot{E}_F(kW)$ | $\dot{E}_P(kW)$ | $\eta_e(\%)$ |
|---|---|---|---|---|
| Seawater pump | 99.81 | 25.61 | 25.57 | 99.81 |
| Condenser | 12.86 | 677.90 | 42.50 | 6.27 |
| Distillation units | 55.11 | 2888.00 | 257.50 | 8.91 |
| Brine pump | 99.72 | 30.14 | 28.32 | 93.96 |
| Product pump | 99.94 | 13.98 | 13.43 | 96.10 |
| TVC | 62.1 | 8612.00 | 5052.00 | 58.66 |
| Total plant | 21.35 | 7723.00 | 239.00 | 3.095 |

**Table 8.** Exergetic variables of the main components of the MSF process.

| Component | $\eta_{e,t}(\%)$ | $\dot{E}_F(kW)$ | $\dot{E}_P(kW)$ | $\eta_e(\%)$ |
|---|---|---|---|---|
| Seawater pump | 99.98 | 97.16 | 97.14 | 99.98 |
| Distillation units | 80.61 | 17,591.00 | 11.74 | 0.07 |
| Brine recycle pump | 99.44 | 258.40 | 255.20 | 98.73 |
| Distillate pump | 99.92 | 20.04 | 19.98 | 99.74 |
| Blow-down pump | 99.98 | 21.82 | 21.80 | 99.93 |
| Brine heater | 99.22 | 3592.00 | 3453.00 | 96.12 |
| Condensate pump | 99.98 | 0.88 | 0.87 | 98.72 |
| Total plant | 17.08 | 3991.00 | 63.12 | 1.58 |

**Table 9.** Exergetic variables of the main components of the DCMD process.

| Component | $\eta_{e,t}(\%)$ | $\dot{E}_F(kW)$ | $\dot{E}_P(kW)$ | $\eta_e(\%)$ |
|---|---|---|---|---|
| Feed pump | 96.13 | 0.03 | 0.001 | 96.12 |
| Permeate pump | 99.90 | 0.03 | 0.002 | 98.85 |
| Distillate pump | 99.98 | 0.02 | 0.004 | 98.28 |
| Heat exchanger | 74.37 | 3.98 | 3.35 | 15.77 |
| Heat recovery | 79.43 | 1.60 | 0.42 | 73.74 |
| DCMD module | 98.28 | 1.87 | 0.12 | 93.60 |
| Solar collector | 95.67 | 4.49 | 0.54 | 88.06 |
| Product tank | 60.58 | 0.42 | 0.16 | 60.58 |
| Total plant | 1.28 | 4.53 | 0.02 | 0.37 |

The pumps used in the MED-TVC process show similar high results according to both exergy efficiency approaches. The feed pump presents the same efficiency results for both cases, because the exergy related to the input stream is zero. In the rest of the cases, the efficiency according to the consumed–produced approach is slightly lower. In the MSF process, the same pattern can be observed regarding the pumps used in this process, with slightly lower results from the consumed–produced approach for all pumps in the system except for the feed pump, where the exergy of the input stream is zero due to dead-state conditions.

For both the MED-TVC and MSF processes, the major difference between the two approaches to exergy efficiency is observed in the distillation units. According to the input–output exergy efficiency approach, all the inlets and outlets of the units are considered.

In the MSF distillation units, the difference in the results of the two approaches is even more pronounced. The extremely small exergy associated with distillate production, compared to the physical exergy supplied to the units for its conversion, leads to a drop in the

exergy efficiency of the distillation units. The disparity in the results of the two approaches in the distillation units is reflected, to a lesser extent, in the total plant efficiency result in both case studies. The exergy efficiency of the MED-TVC plant drops from 21.35% to 3.1%, while in the MSF process, it drops from 17.08% to 1.58%.

Similar results were presented using the consumed–produced approach for MSF systems by Nafey et al. [38] of 1.87% and Mabrouk et al. [39] of 2%. In both cases, only the resources necessary to produce the distillate product of the process were considered as product exergy. An analogous approach, in which the product exergy is defined as the minimum separation work or the minimum amount of exergy input required and the fuel exergy is defined as all the actual input exergy, resulted in slightly higher efficiencies of [37] 4.2% and [40] 5.8%. Al Ghamdi et al. [41] obtained slightly lower results than those previously mentioned for this approach (3%), which may be due to the fact that the chemical exergy produced in the process was not taken into account.

Studies that employed the input–output approach, however, offered very high exergy efficiency values compared to the previous approaches and the results of the present study for an analogous method, with [43] 61.6% and [44] 70%. For the MED process, the results found for the input–output approach were even higher, at [32] 62.7% and [27] 97%.

Carballo et al. [27] also examined a consumed–produced approach similar to that proposed in this study, where the increase in chemical exergy associated with the distillate production is considered as the only product of the process, obtaining an exergy efficiency in this case of 0.2%. An analogous outcome was presented in the study of Khalilzadeh et al. [33], with a process exergy efficiency of 0.1%. Very similar values for the consumed–produced approach to those found in the present study were also presented by Moghimi et al. [29] of 3.1% and García and Gómez et al. [25] of 4.7%. The consumed–produced approach is considered to better fit the thermodynamic behaviour of the distillation units and consequently the overall process where only the distillate is considered as the desired product of the component, obtained from the conversion of physical exergy (mainly thermal) into chemical exergy, which was also observed by Piacentino et al. [26].

In membrane distillation, an exergetic efficiency of 1.29% was achieved for the input–output approach, and it dropped to 0.37% for the fuel–product approach. The main reason is due to the reduced production of product water, which is characteristic of membrane distillation processes and similar to the exergetic efficiencies obtained in other thermal desalination systems. Furthermore, considering exergetic efficiencies only from membrane distillation processes, they are slightly higher compared to some of the studies that appear in the literature [46,47,52,71]. The DCMD module presents high exergetic efficiencies, considering both exergetic efficiency approaches. The efficiency of the DCMD module turned out to be similar to that presented by the VMD module of Miladi et al. [49], reaching exergetic efficiencies greater than 90%, and to Zhu's DCMD module, with an efficiency of 91% [70].

## 6. Conclusions

In this work, a literature review of the application of exergy analysis to the main thermal desalination systems was carried out, with special emphasis on exergy models and exergy efficiency. This contribution also provides an analysis of the two main approaches to energy efficiency: input–output and consumed–produced. For this purpose, three case studies for the multieffect distillation, multistage flash distillation, and membrane distillation processes were studied.

Different exergy models and exergy efficiency approaches applied to the exergy analysis of thermal desalination systems were found in the literature, providing considerable deviations in results and negative exergy, which hinders the standardisation of exergetic analysis.

Equipment involving salt separation and, thus, an increase in chemical exergy plays a fundamental role in the exergy destruction of conventional processes. In the computation of the processes considered in this work, the equipment involved in heat transfer are the main components responsible for exergy destruction, due to the large thermal irreversibilities

produced in them. The low pressure requirements of thermal desalination systems almost eliminate the contribution of pumps to exergy destruction.

In terms of exergy efficiency, input–output approach values for thermal desalination systems present higher results compared to the consumed–produced approach. However, the consumed–produced approach seems to present a better description of the thermodynamic behaviour of thermal desalination processes. The results obtained with the second consumed–produced approach for the MED-TVC, MSF, and DCMD case studies were 3.1%, 1.58%, and 0.37%, respectively, which is in line with the results found in the literature for similar approaches.

Based on the conclusions of this study, we recommend the following:

- The use of the thermodynamic properties of seawater from Sharqawy et al. [23] in exergy analysis, as they seem to better represent the thermodynamic behaviour of seawater.
- The consideration of the consumed–produced exergy approach for analysis, which allows one to focus on the desired product of thermal desalination processes and therefore to better describe the final objective of this type of process.
- The taking into account of the increase in chemical exergy in the definition of the exergy efficiency of the process components in which salt separation takes place.

**Author Contributions:** Conceptualization, N.A.E.K., A.M.B.-M. and N.M.M.; Methodology, N.A.E.K., A.M.B.-M. and N.M.M.; Formal analysis, N.A.E.K.; Investigation, N.A.E.K.; Resources, N.M.M.; Writing—original draft, N.A.E.K.; Writing—review & editing, N.A.E.K., A.M.B.-M. and N.M.M.; Supervision, A.M.B.-M. and N.M.M. All authors have read and agreed to the published version of the manuscript.

**Funding:** This research study has been 85% co-financed by FEDER funds, Interreg MAC Programme 2014–2020, in the framework of the project E5DES (MAC2/1.1a/309).

**Data Availability Statement:** Data is contained within the article.

**Acknowledgments:** This work was supported by the DESAL+ LIVING LAB.

**Conflicts of Interest:** The authors declare no conflicts of interest.

### Nomenclature

| Abbreviations | Description |
| --- | --- |
| DCMD | Direct-contact membrane distillation |
| AGMD | Air-gap membrane distillation |
| VMD | Vacuum membrane distillation |
| MD | Membrane distillation |
| MED | Multieffect distillation |
| MSF | Multistage flash distillation |
| MVC | Mechanical vapour compression |
| TVC | Thermal vapour compression |
| Symbols | Description |
| cp | Specific heat [kJ/kg °C] |
| E | Exergy [kW] |
| e | Specific exergy [kJ/kg] |
| I | Solar radiation [W/m$^2$] |
| N | Number of moles [kmol or mol] |
| *m* | Mass flow rate [kg/s] |
| h | Specific enthalpy [kJ/kg] |
| s | Specific entropy [kJ/kg K] |
| p | Pressure [bar] |
| ppm | Parts per million |
| MW | Molar mass [kg/mol] |
| R | Universal constant of gases [kJ/kmol K] |
| T | Temperature [°C or K] |

| | |
|---|---|
| W | Mechanical work [kJ] |
| $W_{min}$ | Minimum work of separation [kW] |
| $m$ | Mass flow rate [kg/s] |
| $u$ | Chemical potential [kJ/kmol] |
| $x$ | Mole fraction |
| $w$ | Mass fraction |
| $\eta_e$ | Exergy efficiency [%] |
| $\varphi$ | Dissociation factor of the salts |
| Subscripts | Description |
| $i$ | Related to chemical species |
| 0 | Dead state |
| D | Destruction |
| F | Fuel |
| L | Loss |
| P | Product |
| no dissoc | Not accounting for the ionic dissociation of salts |
| in | Input |
| out | Output |
| hot | Hot stream |
| cold | Cold stream |
| sol | Solar |
| t | Total |
| s | Salt, saline water |
| w | Pure water |
| Superscripts | Description |
| CH | Chemical |
| KN | Kinetic |
| PH | Physical |
| PT | Potential |

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
