# Peer review of "Definition of Exergetic Efficiency in the Main and Emerging Thermal Desalination Technologies: A Proposal"

_water, doi:10.3390/w16091254_

Round 1
Reviewer 1 Report
Comments and Suggestions for Authors
The article is devoted to an important and topical issue related to the application of exergy analysis to desalination processes.
The exergetic method of thermodynamic analysis is a relatively new branch of thermodynamics; it is based on the application of the concept of exergy to the study of technical processes and relies on the use of thermodynamic potentials to analyze energy conversion processes in various systems. Properly selected thermodynamic potentials have the extremely important property of giving the value of work (both mechanical and any other type, for example electrical) under certain conditions. Using this fundamental property of potentials, it is possible to assess the performance of matter and energy flows at any point in the system under consideration, regardless of its type, structure and complexity. This method is used in all areas of thermal power engineering and heating engineering. The use of exergy analysis is very effective in the study of chemical-technological systems based on balances that have a large number of energy sources and sinks. Using this approach, the problems of creating energetically closed chemical production facilities are solved, since it is possible to assess both internal and external losses, as well as the potential of energy flows. The method is widely used in the calculation of heat exchange systems, comparative assessment of various methods for separating multicomponent mixtures, analysis of chemical technological systems, etc.
Based on the foregoing, the presented work is relevant and will be of interest to the reader in the area under consideration.
However, a number of comments about the article should be clarified:
1. The introduction presents a fairly good literature review on the subject under study. . In the introduction, it is necessary to explain in more detail what energy intensity the thermal processes of multi-stage distillation have.
2. The application of exergy analysis for a solar installation seems interesting [23]. It should be clarified what parameters of the solar installation were considered and can exergy analysis be applied to the analysis of other renewable energy sources?
3. Tables 1-3 could provide numerical values of the exergy efficiency factor.
4. Not all numerical designations in Figures 1-3 are deciphered.
5. Has the scheme shown in Figure 3 been considered for a wider range of input parameters considered?
6. The work should explain what software was used to process the results of experimental studies.
7. According to the data given in tables 6-8, it would be possible to carry out a correlation and regression analysis with specific mathematical models that allow the calculation and prediction of the output values under consideration.
8. A generalized research methodology should be presented for which a corresponding patent could be obtained.
9. In the conclusions, it would be possible to dwell in more detail on the testing of the results obtained on specific objects and the prospects for further research on the topic under consideration.
Reviewer 2 Report
Comments and Suggestions for Authors
In this paper, the authors present an overview of the application of exergetic analysis to the main thermal desalination systems, with a focus on exergetic models and energy efficiency. However, in the course ofreading the work, I noticed several flaws:
1. In my opinion, the authors should indicate in the review more data on the latest relevant research related to liquid desalination technologies.
2. I think that the authors need to improve the quality of the diagrams. In my opinion, it is worth increasing the font of the titles.
3. The authors should also improve the quality of figures 1, 2, 3. Increase the font of the names of the elements. Replace the light colors of the circuit lines with clearer ones.
4. In my opinion, the authors should indicate the relationship between the theoretical method of analysis and practical data from other works.
Reviewer 3 Report
Comments and Suggestions for Authors
Dear Authors,
My understanding, the main aim of the article is to address the lack of standardisation in defining exergetic efficiency, particularly in the context of desalination processes. However, it remains unclear to me whether this goal has been achieved and the methods employed to achieve it are not fully elucidated.
Please address the following questions/concerns and incorporate into the context of the article:
I found many English mistakes, please revise the language of the article.
Add a methodology section addressing the method used in this article to achieve its ultimate goal.
The article lacks a clear discussion of the specific methodologies or criteria used to compare and critically review the different approaches to formulating exergetic efficiency.
Section 4 titled Methodology is not representing methodology, I think ether become part of the literature review or up to you to title it differently.
I don’t understand how the results is contributing to the standarisation?
Add a flow diagram outlining the progression of the article's content, starting with the aim, followed by methodology, approach, results, and discussion, all contributing to the ultimate goal of standardization.
Comments on the Quality of English LanguageModerate English editing is required, notice multiple mistakes.
Round 2
Reviewer 3 Report
Comments and Suggestions for Authors
I believe the authors have addressed most of the reviewers' comments.
Comments on the Quality of English LanguageEnglish language is fine.